# Monuments and Monumentality in a Changing Socio-Political Landscape: A View from Udaypura

**Rafia Khan**

Institute of Management, Nirma University, Ahmedabad 382481, India; khanrafia.niz@gmail.com

**Abstract:** This work intends to explore the nature of socio-political change in historical periods usually referred to as interregnal which, for the purposes of this paper, is defined as a period of discontinuity or gap in political and social organization. It traces the survival of a historical monument through two interregnal centuries of medieval Indian sub-continental history (11th–12th and 14th) to argue that modern historiographical templates which study these periods as precursors or remnants of succeeding and preceding centuries, respectively, do not sufficiently explore the socio-political possibilities innate in these periods of distributed political agency. In the context of Indian history, while historians have focused on the confrontational aspect of Hindu–Muslim polities or communities in interregnal centuries, I suggest that these periods provided fertile ground for political innovation and negotiation, thus breaking the confrontational stasis usually associated with regnal centuries.

**Keywords:** Indian subcontinent; socio-cultural history; monuments; Delhi Sultanate and the Paramaras; interregnum centuries





## 1. Introduction

For the modern historian of the Indian subcontinent, certain periods represent an elusive temporality. The 8th–12th centuries CE, which could neither be aligned linearly with the glorious kingdoms of the ancient, nor shown to have directly contributed to the advent of Turk hordes which dominated the higher echelons of political power in medieval India, was one such period. Being at once 'post-ancient' and 'pre-medieval', the historiography of this period in the 1990s was subject to a critical review by B.D. Chattopadhyay, who argued for the de-linking of the developments of this period as remnants or precursors of preceding and succeeding centuries. He spoke of independent, local state-formation at regional and sub-regional levels coinciding with the flourish of multiple intense but regional economies, giving birth to a large number of widely distributed suburban centres of religion, culture, and politics [1].

This is an important lesson for any student of 'interregnum' centuries, such as the 8th–12th, the 14th–15th, or the 18th centuries which are easily characterised as 'dark ages', periods of dynastic gaps or decentralisation, as opposed to periods of integrated, interconnected macro-regional empires like those of the Guptas, the Delhi Sultans, or the Mughals. These were also identified as periods of the slow demise of classicism in literary and architectural production to the advantage of untraceable developments from 'below'. Yet, these historiographical templates are often challenged by the coexistence of unintegrated local polities making competitive, universal claims of rulership within large empires. We find *cakravartins* with nominal, local, and sub-regional authority and emerging, local elite groups alluding to vastly spread inter-regional political networks around themselves. It is also not surprising that these binaries themselves reflect a spatial and temporal simplicity, a sort of short-circuiting of socio-political processes which seems farcical in the face of historical realities.

Samira Sheikh and Francesca Orsini made an argument similar to Chattopadhyay's for the late 14th and 15th century. They questioned the characterisation of the late

14th and 15th century as the 'twilight' of the Delhi Sultanate. In their attempt to pull this period out of the abyss of political insignificance and cultural vapidity which it had fallen into in modern historiographical works, they highlighted the innovative cultural and literary mediations of 15th century courts, such as the adoption of vernaculars to courtly standards even as classical literary forms and languages continued to be patronised. Another section focusing on political diversity, drew connections between new forms of literature and the emerging social groups which patronised them as works of self-expression and self-identification so that they became ways of 'producing one's own history or inscribing oneself in larger histories' [2].

Yet, as Pankaj Jha correctly points out in his work *A Political History of Literature*, their effort at registering the cultural and literary efflorescence of the 15th century as an affront to the traditional historiographical template subscribed to another widely held historiographical assumption, that of the primacy of political history vis-a-vis other spheres of historical activity, at least in the title of their work, *After Timur Left*. With his own work on Vidyapati and the copious literary production of the small kingdom of Milthila in the 15th century, Jha showed that 'any easy correspondence between the 'size' of a kingdom and the political imagination of its literary elite would be misleading' [3]. In simple words, tying the prosperity of socio-cultural realms of expression to the presence of a typical state may be a historiographical condition, but not a historical one.

Such historiographical templates also find it rational to ascribe monuments and monumental activity to political prosperity and stability. In this context, the proliferation of cultural objects such as built monuments in 'interregnum' centuries has not been widely recognised or explained. There are two reasons for this. First, the apparently direct links between 'political difficulties' and the 'little time this leaves for constructional activity' has led to the oblivion of the fact that constructional activity was not always a by-product or result of political developments but rather, often a means of establishing or reinforcing new socio-political relations. Secondly, the assumed dyad that links 'poor construction' to sparse financial resources does not recognise architectural reuse and reconstruction as meaningful architectural activity. One way to address this gap as suggested by Martin Furholt and Doris Mischka is to recognise 'the crucial role of monuments in periods of massive social and political change in the shaping of new, previously unprecedented social systems' [4]. This has significant ramifications for our understanding of heightened constructional activity in 'interregnum' centuries.

## 2. The Udayeshwara Temple

The Udayeshwara temple which lies in the modern Udaipur village of the Ganjbasoda district of Madhya Pradesh today is one such unique example which presents in its monumentality and inherent classicism of architectural style, a moment of architectural and political effervescence in a period of regional fragmentation and political oblivion for the Paramara rulers of Malwa. King Udayaditya (c. 1070–1093) attempted to restore the glory of his dynasty years after the demise of his famed, larger than life predecessor King Bhoja (1000–1055) by commissioning a monument equal in magnitude to the incomplete Bhojpur group of monuments planned and commissioned by his powerful predecessor.

As Doria Tichit writes, the Udayeshwara temple artistically represented not just architectural emulation of the 10th century *sekhari* mode of temple architecture, but also gives evidence of a climate of architectural effervescence and invention by being one of the earliest and finest examples of the *bhumija* mode of temple architecture. Tichit not only attests to its royal foundation but observes that as an early example of a recently developing architectural style, 'some clumsiness could've been expected in its realisation' but against those expectations, the Udayeshwara temple is remarkably fine in technique and proportion and indicates a 'mastery of the newly elaborated architectural mode' [5].

Udayaditya (c. 1070–1093) ascended to the throne at a critical period. The Chalukyas and the Kalchuri rulers had wreaked immense dynastic misfortune upon the Paramaras of Malwa towards the end of Bhoja's rule. Indeed, these developments seem to have left

Bhoja devastated, who ultimately died leaving his empire in the throes of an inter-regional and inter-dynastic conflict. His immediate successor regained the throne with the help of his political allies, but it remained an uneasy political alliance and eventually, cost him his life. In these circumstances, a Paramara inscription informs us, 'arose king Udayaditya, like another son, destroying the dense darkness, his powerful foes with the column of rays issuing from the strong sword' [6]. He backed his victories with multiple matrimonial alliances and articulated his political and ritualistic authority over the subject population through the religious cult of the Udaleshwar. Udayaditya was clearly the eponym for the deity visualised in the typical cult object of the mahalingam which sits in the sanctum sanctorum of the Udayeshwara temple. The temple and its deity's direct association with Udayaditya also added a political sacrality to the space, which was precisely the reason it had to be dismantled when a new political order set in [7].

When in the 4th and 5th centuries of the second millennium, the military retinue and political culture of the Sultanate/s came to Malwa, the Udayeshwara temple's monumentality underwent that typical desecration, reuse, and restructuring programme, which was a part of the Delhi Sultanate's military conquest of a region. As the political empire of Muhammad bin Tughluq (1325–1351) extended across the regions which constituted the dakshinapatha, the governor of Chanderi, Izz-ud-Dīn Bantānī had by 1338–1339 A.D. managed to push the extent of the Chanderi province up to hundred kilometers to the south-west of the provincial capital at Chanderi, till Udayapura. Bantani appointed a sub-provincial representative in the vicinity of Udayapura and it seems to have fallen to this sub-provincial officer to desecrate the monumentality of the Udayeshwara to indicate the disruption of the old political order and reflect the new political realities of the region [8,9].

Political (and not religious) acts of desecration and/or restructuring built monuments of religious-cultural and socio-political significance were undertaken in different parts of the subcontinent in these centuries. Many sites in Central India, such as the Mattamayura Saiva mathas at Kadwaha near Chanderi, the Nandi temple at Bhilsa (Raisen), as well as countless consecrated Jain idol complexes in the modern parts of Bundelkhand, underwent different degrees of desecration, indicating that this was a common political activity in this period. Scholars have revealed the predictability of this desecration and restructuring program, inasmuch that as a signifier of political conquest and as a platform for establishing new socio-political relations and hierarchies, it constituted fairly predictable and oft-repeated stages. Most typically, temples considered essential to the constitution of enemy authority were destroyed. Occasionally, these were converted into mosques [10]. More rarely, the temple deity was abducted and brought to the conqueror's capital. The political violence at Udayeshwara is unique when compared to these typical patterns of temple desecration.

## 3. The Political Restructuring of the Udayeshwara Temple

The Udayeshwara temple comprises a heavily sculpted central temple. The temple comprises of the shrine (mulaprasada) and the assembly hall (mandapa) with a rich and dense iconographic program adorning the superstructure (Figure 1).

This temple is situated at the centre of a temple complex which also includes independent *vedis* (reading rooms) or small shrines in all cardinal directions around it. When J.D. Berglar, assistant to the first director-general of the Archaeology Survey of India (ASI), Alexander Cunningham, visited Udayapur in 1871–1872, he wrote about the Udayeshwara temple complex which consisted of a mosque 'at the back of the temple'. (Figure 2) He also referred to the makeshift inscribed archways (Figure 3) which provided a ritual entry into the mosque, but which could be accessed from any other place on the complex as well. Later, Alexander Cunningham reported that 'the north-west corner temple and the western bedi *(vedi)* were knocked down in the time of Muhammad-bin-Tughluq and a *masjid* was erected in their place [11,12].

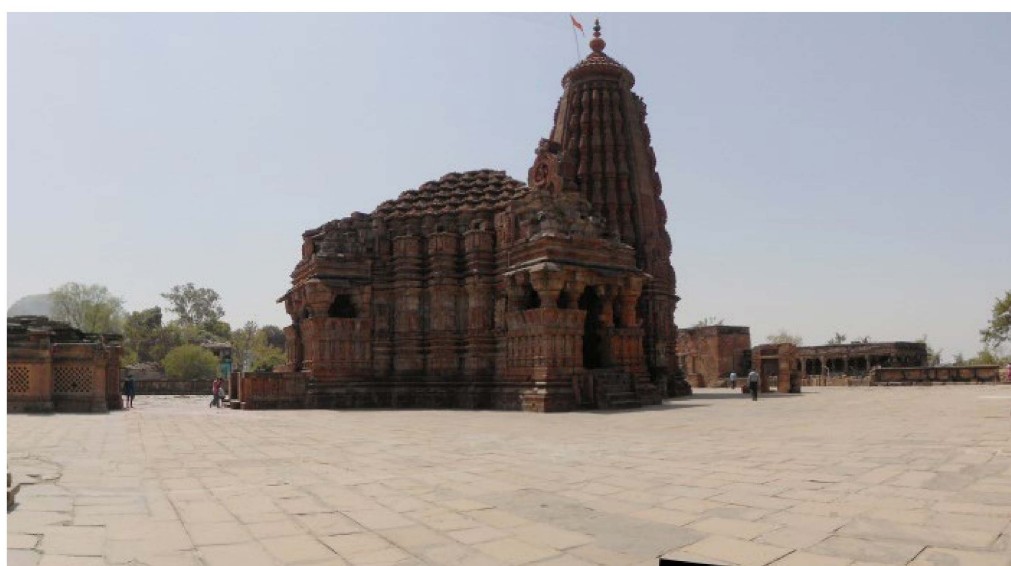

**Figure 1.** The central structure with the mulaprasada and the mandapa at the Udayeshwara temple ('Udaipur_panorama' by Deepesh9929 is licensed under CC-BY-SA-3.0).

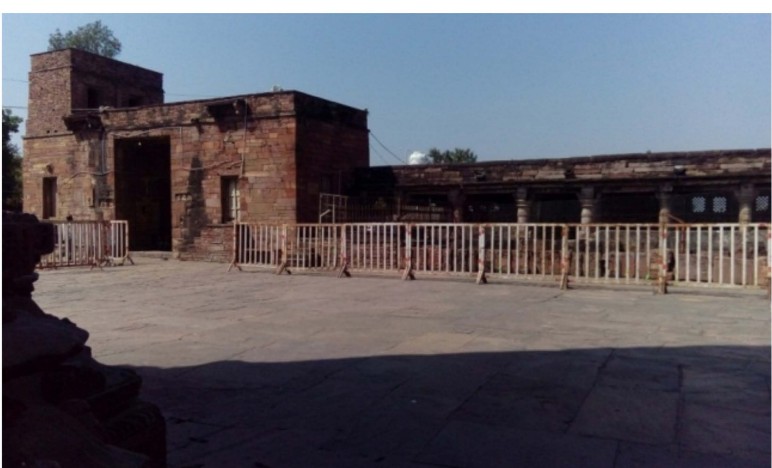

**Figure 2.** The makeshift trabeate mosque at the back of the temple lies behind the barricade. (Image from author's personal collection).

Apart from the monument itself, two parts of a Sultanate inscription dated to 737/1336–1337 and 739/1338–1339 found at the site attest to this 14th century restructuring. The inscriptions stay silent on the matter of temple desecration but simply credit the commission of a mosque to Azam Malik, the *sarjandar-i-khass* (commander of the special forces) of the Sultanate court. In political terms, these episodes must be seen in Richard Eaton's words as 'a normal means of decoupling a Hindu king's legitimate authority from his kingdom, and specifically, of decoupling that former king from the image of the state deity that was publicly understood as protecting the king and his deity' Eaton, p. 73).

Yet, this desecration-restructuring program was unconventional as it did not seem to desire a complete annihilation of the site or the structure. It is evident that the central temple which was the cynosure of all religious and political activities, carried all the pilgrimage inscriptions, housed the *mahalingam*, and dominated the entire temple complex with its massive iconographic programme was not under attack. The *mulaprasada* or the shrine, the most sacred space of the temple which houses the cult object, opened towards the east to let the light of the rising sun enter inside the temple and bathe the cult object, i.e., the mahalingam. The Sultanate restructuring did not disturb this cosmological arrangement and left the path for ingress and egress open for sunlight and devotees, so that only a

'minor' temple or what was also referred to as the attendant shrine, 'at the back of the temple' was brought down. A more modern reading of the present architectural plan of the structure by Doria Tichit also confirms that 'the design of the ensemble has been modified by the construction of a mosque on the west part of the platform and of two arches at the back of the temple' (Tichit, pp. 46 and 101).

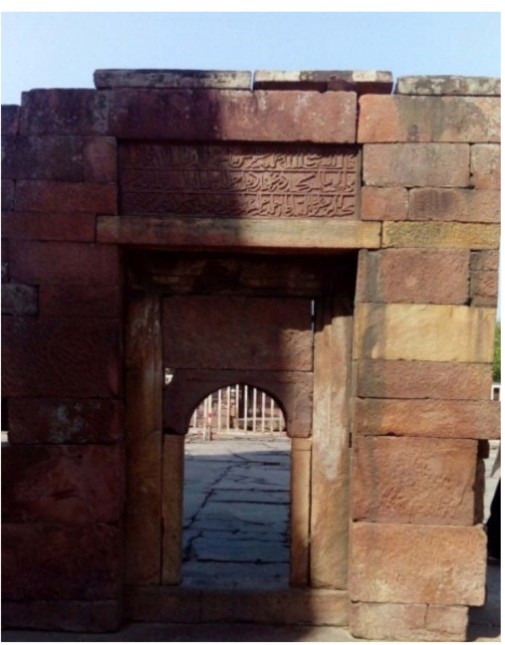

**Figure 3.** The inscribed archway which provided ceremonial entry to the mosque (Image from author's personal collection).

Was this political restructuring of a monument while keeping its monumentality intact unique? The unconventional nature of this temple desecration becomes clearer when we contrast it with other patterns of desecration and restructuring in medieval central or northern India. In the mid-12th century, the Rudramahalaya Saiva complex in Siddhpur, Northern Gujarat underwent a programme of political iconoclasm wherein 'the complex's principal temple was completely dismantled, its minor shrines were left standing and uneffaced.' [13]. These were then reused to form parts of an Islamic religious structure. A late 17th century example from the same region was the destruction of the Bijamandal temple complex by Aurangzeb, considered on all accounts to have been a destructive force. Coming back to the 14th century, Tamara Sears concludes her study of the political restructuring of the Saiva matha complex at Kadwaha with the understanding that 'the tenth century Siva temple built next to the monastery may have been at one point completely covered, and that the platform upon which the mosque currently stands once possibly extended farther east to cover the entirety of the earlier temple' [14]. These instances indicate that the desecration at Udayeshwara was unique for those times in preserving the monumentality of the built object, even in the face of the complex's destruction.

On the other hand, the desecration at Udayeshwara shared motive with other such acts of political desecration. Most acts of temple desecration and restructuring were strategic attempts to articulate a shift in political power. These announced the presence of new political rulers. As these were not given to religious iconoclasm or the spread of Islam, pre-existing patterns of religious devotion and ritual continued to survive around these politico-religious structures. This was true for the Udayeshwara temple, too. The absence of any intense desire to change or overpower established religious beliefs was reflected in the continuity of pilgrimage traditions which seem to have continued in these years of desecration as well. Thus, pilgrim inscriptions dated V.S. 1394/1338 inform us that Ratana with his wife and son came to the temple for a visit to Udalēsvara Deva. Another of the same year refers to the yatra festival of the god Udalēsvara. That the perpetrated 'violence'

on the monument complex was not a reason to drive the devotees and pilgrims away abruptly, and that the monument was open to pilgrimage processions and related rituals, leads one to question the assumed 'violent' nature of such acts of political iconoclasm. Indeed, the inscriptions at the temple suggest that pilgrims continued to visit the temple site for a large part of the 14th and 15th centuries and dated inscriptions till 1447 are available to attest this [15].

This indicates that the religious and devotional structures woven around the monument or its monumentality did not alter significantly in the reign of the Delhi and the Malwa Sultans.

## 4. Monumentality in Interregnum Centuries

In the context of the aforementioned events, the commissioning of the Udayeshwara temple in the 11th century and the desecration it experienced in the 14th century are unique developments belonging to 'interregnum' centuries.

The temple's construction in a period of political discontinuity was an attempt at the reconstruction of Paramara prestige which had undergone some degree of decline after Bhoja's reign. What made this enterprise different from Bhoja's intense construction activity, however, was that the monumentality of the Udayeshwara served not as an expression or symbol of, but rather as a tool of establishing Paramara revival. This alone explains the fact that the temple was completed and consecrated within the first 10 years of Udayaditya's long 23-year rule. At a primary level, this was achieved by commissioning a monument of public utility. At a more complex level, the intentionality of commissioning a temple with more non-functional elements, such as the temple's rich and complex iconographic program, its massive scale, or the multiplicity of its entrances, than functional elements indicate that the purpose of commissioning the temple was more than to merely serve the quotidian spiritual needs of the people. In fact, multiple groups of people of differing statuses would have participated in the creation of the temple's monumentality. Not only that, in this period itself, the temple and the deity seem to have been at the centre of a ritualistic programme which included pilgrimage, installation of religious objects such as standard flags, and inscriptional scribbling as some of its more visible elements. Participation in these ceremonies which revolved around the eponymous king and deity shaped the identity and relation of the worshipping population as subjects vis-a-vis their temporal and spiritual ruler. In this sense, the monument and its monumentality were an instrument to establish and visualise the Paramara political hierarchy which emanated in the figure of Udaleshwar, and through him, in Udayaditya in a period of massive socio-political change.

The desecration of the Udayeshwara temple does not necessarily belong to an interregnal century. The political clout of Muhammad-bin-Tughluq and the Delhi Sultanate was intact throughout the 14th century. Political communication was ensured through multiple informal and formal channels. A considerable standardization of Sultanate presence in provincial and sub-provincial settings was also ensured through a migrating administrative bureaucracy. Tughluq's rule, however, was marred by a spate of rebellions. There is much to indicate that a considerable section of the rebelling Sultanate ruling elite was affecting local and provincial state-formation across the Sultanate dominions in the 14th century [16].

Even though the administrative hierarchy of the Sultanate was visible in the mosque inscription at Udayapur, the Sultanate representative on ground seems to have enjoyed enough room for independent political manoeuvring. This explains the distinct desecration pattern at Udayeshwara.

Local folklore would explain this alternative pattern of desecration by resorting to stories of divine interference. Richard Eaton understands that desecration was only reserved to active centre of politics in the 14th century and that this was the reason that the temples at Khajuraho escaped desecration. This does not apply to Udayeshwara because it was held by a local Paramara chief in the early 14th century which was precisely the reason it could not escape desecration.

What was unique was the pattern of desecration wherein the monumentality of the structure is preserved even in the face of its desecration. It may have been the result of an alternative strategy of political communication and articulation, where the monumentality is preserved as a reminder of the benevolence and protection of the Sultanate. Considering the Sultanate officers' intention was never to change the devotional patterns which emerged around the temple, and therefore, visitors and pilgrims continued to visit the temple till mid-15th century, the monumentality of the Udayeshwara temple may have stood to these pilgrims as proof of the protection of the Sultanate. If the local populace at Udaypura had agreed to pay the jizyah (a religious tax on non-Muslims), then they would have acquired the zimmi status which would have made the state the guardian of their life and property, thus securing the temple's monumentality. Thus, the semi-destructive state of the temple could be a sign of negotiation between two political communities which were confronting each other because of the changing socio-political presence of the Sultanate from conquerors to rulers or administrators. On the other hand, the preservation of the monumentality of the complex might also be a symbol of the Sultanate's power in the sense that while many other temples were desecrated at this time, this one stood as the provincial Sultanate administration allowed it to. We do not have much insight into the socio-political interaction between the local populace and the provincial administration at Udayapur but it is not unreasonable to suggest that these relations may have been more cooperative here than in some of the other Sultanate provinces.

Whatever may have been the reason for this pattern of selective desecration, the fact that it was affected by a sub-provincial level Sultanate officer at a time when the Tughluq Sultan's authority was waning at the hands of his own ruling class is critical. This is not to suggest that the Sultanate representative or local governor at Udayapur was unmoored from the 'Islamicate' culture to which the Sultanate subscribed, or that the distance from the capital, in itself, would lead to the adoption of alternative strategies of political negotiation and communication. It is to suggest that the Sultanate and its administration was no monolith which could be reproduced in discrete provincial and sub-provincial settings. Alternatives or amendments to widely accepted patterns of conquest and state-formation developed as the Sultanate's administrative paraphernalia spread into the subcontinent and central control weakened. It is possible that the interregnal nature of Udayapura's settings encouraged such political innovations.

## 5. Conclusions

The events at Udayeshwara change our understanding of the monolithic nature of change and historians' characterisation of 'interregnum' centuries as unstable or chaotic. Interregnum centuries are unique because they provide insights into the interactions between pre-existing and emerging historical elements. The persistence of pre-existing socio-political, cultural, and religious networks in the face of a markedly visible and shifting political paradigm suggests that any sort of direct correspondence between the two belies the complex historical reality of these centuries. The fact that there were different versions of the Sultanate iconoclastic program, and not one established or predictable programme of desecration, indicates the exploratory character of socio-political activities in interregnum centuries. This is in no way to suggest that non-interregnum centuries have a set pattern which remains static or uniform but that the decentralisation of power and resources brings more agency to historical actors on the ground. It is also to argue that the long-drawn and continuing process of socio-political exploration, the ultimate shaping and establishment of political choices, and their eventual contribution to state formation is difficult to recover from hegemonic and regime-centred historical sources. The 8th–12th centuries, the 14th–15th centuries, and the 18th century provide novel opportunities to observe and study these political alternations and innovations.

**Funding:** This research received no external funding.

**Conflicts of Interest:** The author declares no conflict of interest.

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
