# Peer review of "Monuments and Monumentality in a Changing Socio-Political Landscape: A View from Udaypura"

_2409-9252, doi:10.3390/histories1040024_

Round 1
Reviewer 1 Report
The author must cite the name of the authors that are given direct quotes in the text. The reader should not have to dig into the footnotes to learn who is speaking in such quotes. Examples:
Lines 55-56, 74-78, 103-04, 156-59, 190-93.
Similarly, the author must identify “Berglar” cited in line 146,
The author must also locate the site of this temple. Simply saying that it is in Ganjbasoda District in M.P. is not sufficiently precise. WHERE in that district is the temple?
Author Response
Dear sir
Thank you for your comments. These were helpful and all changes asked for have been affected. I believe that there is some scope for improvement in terms of the argument that I am making. However, the time allotted to me is too little to allow me to sharpen the argument. If I could seek another week's extension, I am confident of bringing greater sharpness to my argument.
Reviewer 2 Report
The interregnum periods in the history of the Indian subcontinent create many problems in historical research. Monuments built or rebuilt at that time represent a diverse political situation. It is also right to ask the question: are monuments only a secondary form of political life, or are they sometimes its main tool? The history of the Udayeshwara temple in Madhya Pradesh is a good example of the building's involvement in policy making, both in consolidating the current power and in humiliating previous rulers. The destruction of other temples is an additional argument for the thesis about the important role of architecture in political life. In general, it can therefore be concluded that the periods of "historical discontinuities" are densely filled with events that also arouse political emotions today.
Author Response
Dear sir
Thank you for your comments. I appreciate the summation which you've provided and would want to think further on the questions posed. However, as these are questions of a theoretical and complex nature and a conclusion can be arrived at only after great deliberation, and I being granted only a week for submission, have only summarily managed to include them in the present work.